# Memetic Computing for Healthcare Resource Management: An Integrated Planning for Operating Theaters, Rooms, and Continuity of Care

**Javier Almenara-Herrera, Eduardo Segredo, Gara Miranda**
Departamento de Ingeniería Informática y de Sistemas
Universidad de La Laguna
San Cristóbal de La Laguna
Spain
(javier.almenara.12|esegredo|gmiranda)@ull.edu.es

## Abstract

This paper addresses integrated planning in the healthcare sector by formulating the Integrated Healthcare Timetabling Problem (IHTP) as defined in the IHTC competition. The problem integrates the assignment of operating theaters, hospital admission planning, and the allocation of nurses to patient rooms. Although the literature has traditionally treated these subproblems in isolation, our work emphasizes an integrated approach that more accurately reflects the real-world operations of hospitals. We hypothesize that the application of a memetic algorithm will yield high-quality solutions that significantly enhance operational efficiency and resource management in healthcare facilities.

## 1 Motivation

The healthcare system is essential in modern society, as it directly impacts the quality of life and well-being of people [13, 16]. An increase in the demand for medical care is expected, driven by demographic changes such as longer life expectancy, improved quality of life, and the growth of the elderly population in developed countries [7, 12]. This scenario has led to a considerable increase in healthcare care expenditure in recent years [7, 10]. Faced with increasing costs, healthcare providers are compelled to research and implement innovative methods to contain these expenses, which in turn requires a more effective and efficient organization of processes [7].

Planning the optimal use of healthcare resources presents a complex challenge given the multiple factors that complicate decision making at all levels. Among these challenges are the internal fragmentation of large institutions —for example, the organization of hospitals into numerous autonomous units [13, 14, 15]—, conflicting objectives, and limited collaboration among key stakeholders such as physicians, nurses, and administrators [5]. Furthermore, the lack of essential data for proper planning and oversight [1] and the high daily variability in service demand [4] underscore the need for sophisticated planning methods that facilitate more accurate decisions and efficient use of available resources [13].

Since the 1950s, the application of Operations Research (OR) and Management Science (MS) has provided analytical frameworks that enable the efficient administration and planning of limited resources, driving significant advances in the healthcare sector [6]. Although focusing on the organization of individual resources —-such as operating rooms, beds, or specific personnel profiles— can improve efficiency, this strategy often overlooks the complex interrelations among different resources and divisions within organizations. This approach may lead to decisions that do not optimize the overall functioning of the system, a situation particularly evident in hospitals that, despite having

XVI XVI Congreso Español de Metaheurísticas, Algoritmos Evolutivos y Bioinspirados (maeb 2025).

state-of-the-art technology and high clinical specialization, experience limited coordination due to departmental autonomy [7, 14]. Therefore, there is a recognized need to develop OR/MS models that allow for an integrated planning of multiple resources [6, 8, 15]. In this regard, vertical integration —oriented towards the joint management of various resources— emerges as a promising strategy, supported by an increasing number of studies advocating for integrated planning approaches [2, 6, 15].

In this context, the *Integrated Healthcare Timetabling Competition 2024*[1] is introduced as an initiative aimed at promoting research on integrated optimization problems in healthcare. The competition focuses on jointly addressing three critical challenges: the scheduling of operations, the planning of hospital admissions, and the assignment of nurses to patient rooms. This proposal aligns with the need to develop integrated planning strategies that overcome fragmentation and enhance coordination within health systems. IHTC 2024 provides a comprehensive framework by defining a detailed problem formulation based on hard and soft constraints, along with a set of test instances and a validation tool to assess the feasibility and quality of proposed solutions. This challenge not only represents an academic exercise, but also serves as a platform to generate solutions that are applicable to the real-world challenges faced by the healthcare sector.

## 2   Working hypotheses

The integrated problem proposed in the competition is decomposed into three NP-hard subproblems and involves critical decision-making processes, including:

1. Determining the admission date for each patient or postponing their admission to the next scheduling period.
2. Selecting the appropriate operating room for each admitted patient.
3. Assigning a room for the entire duration of the patient's stay.
4. Designating the corresponding nurse for each room in each shift of the scheduling period.

The model is governed by an objective function to be minimized, which integrates a series of adjustable weights associated with soft constraints, along with a set of hard constraints essential to ensure solution feasibility. In particular, hard constraints ensure, for example, compliance with capacity limits in patient rooms and operating theaters, the mandatory admission of required patients within the established period, and the limited availability of human and technical resources.

At the same time, the soft constraints reflected in the objective function aim to minimize factors such as delays in patient admissions, disruptions in continuity of care, deviations from the required nurse skill levels, and nurse workload overload, as well as the unnecessary opening of operating theaters and changes of surgeon between different operating theaters.

It is hypothesized that a memetic algorithm can be applied to tackle the integrated healthcare scheduling problem to improve operational efficiency. Memetic algorithms have shown promising results in a wide variety of complex optimization problems [3, 9], suggesting their potential to bring about substantial improvements in decision making within the healthcare sector.

## 3   Objectives

Throughout the design and development of this proposal, multiple objectives are pursued to contribute to the field of integrated optimization in healthcare systems:

- **Formulation of a holistic model:** Formulate a model that jointly addresses the three subproblems defined in the competition —scheduling of surgical cases, planning hospital admissions, and assignment of nurses to patient rooms— thereby overcoming the limitations of fragmented approaches and achieving a global coordinated view of resource management.
- **Design, implementation, and tuning of a memetic algorithm:** Design and implement an algorithm that combines a global search procedure with local refinement, capable of efficiently exploring the vast solution space and finding high-quality solutions.

---

[1] https://ihtc2024.github.io

- **Establishment of a rigorous evaluation framework:** Define and apply precise metrics to quantify operational efficiency, costs, and the overall quality of solutions, aligned with the objectives of the competition, to facilitate exhaustive comparative analysis.
- **Empirical assessment of the metaheuristic's performance:** Conduct a comparative study to determine whether the application of the memetic algorithm offers significant improvements in integrated planning of the healthcare system compared to the solutions provided in the competition.
- **Exploration of complementary aspects:** Evaluate the sensitivity of the algorithm to parameter variations, analyzing the scalability of the model in more complex scenarios and considering its application in a local healthcare system.

## 4 Description of the methodology

The methodology adopted in this study is based on a systematic approach that ranges from a theoretical review to experimental implementation and evaluation, following these phases:

### 4.1 Literature review and problem analysis

An exhaustive review of the specialized literature on optimization applied to healthcare planning will be conducted. This analysis will help identify the main limitations of existing approaches and justify the need for an integrated strategy that jointly addresses the scheduling of surgical cases, the planning of hospital admissions, and the assignment of nurses to patient rooms.

### 4.2 Problem formulation and modeling

The problem will be formalized by defining the variables, constraints, and objectives that govern the integrated planning. The resulting model will be based on representative instances from IHTC 2024, ensuring the relevance and applicability of the proposal.

### 4.3 Development and implementation of the memetic algorithm

The cornerstone of the methodology consists of designing, implementing and fine-tuning a memetic approach. This process is structured in the following steps:

- **Solution encoding:** An adequate representation of the integrated problem is sought, which is crucial given the large number of variables and factors to consider. Therefore, selecting a suitable encoding constitutes a significant challenge.
- **Initial population generation:** The algorithm population is initialized using random generation techniques, which helps ensure sufficient initial diversity to favor the exploration of the solution space.
- **Constructive heuristic:** A constructive heuristic is used to generate solutions from the encoding of the problem. This heuristic is applied at various stages of the process: first, it seeks to minimize the number of operating theaters opened by reducing their daily available time; additionally, nurses are sorted and sequentially assigned to those rooms that contain patients with higher care requirements.
- **Definition of evolutionary operators:** Operators such as selection, crossover, and mutation are established to efficiently explore and exploit the solution space. In particular, generational selection with elitism, a PMX crossover operator adapted to the problem encoding, and a mutation applied to admission days assigned to patients will be used.
- **Local search:** Local search techniques will be implemented to further refine the solutions obtained by adjusting parameters that improve final quality and prevent premature convergence.

### 4.4 Evaluation and analysis of the solutions

A series of experiments will be designed using predefined instances, in which the performance of the solutions will be evaluated using metrics that quantify operational efficiency, costs, and overall

quality. The algorithm output will comply with the competition's requirements and will be validated by a system that checks the absence of hard restrictions violations, as well as by calculating the cost of the objective function: identifying the number of infringements for each component and their corresponding weights. Additionally, a complementary evaluation will be carried out using the test files provided in the competition.

### 4.5 Documentation and conclusions

Finally, the findings will be systematized and analyzed. Detailed documentation of the experimental process and a critical discussion of the results will allow drawing well-founded conclusions regarding the effectiveness of the evolutionary approach in the integrated planning of the healthcare system, as well as its potential extrapolation to local healthcare systems, thus transcending the initially posed problem.

## Acknowledgment

This work is part of the project PID2023-152614OB-I00 funded by MI-CIU/AEI/10.13039/501100011033 and by "ERDF/EU".

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
