# OpenReview forum: "Memetic Computing for Healthcare Resource Management: An Integrated Planning for Operating Theaters, Rooms, and Continuity of Care"
_MAEB/2025/Projects_Track — MAEB 2025 Proyectos_

### Official Review · Reviewer_KP2d · 2025-03-17
**Memetic Computing for Healthcare Resource Management: An Integrated Planning for Operating Theaters, Rooms, and Continuity of Care**

**Rating:** 2
**Confidence:** 4

**Review:**

Introduction

This work proposes a memetic algorithm for a timetabling problem defined in the IHTC competition (Integrated Healthcare Timetabling Competition). The problem integrates the assignment of operating theaters, hospital admission planning, and the allocation of nurses to patient rooms. The integrated problem proposed in the competition is decomposed into three NP-hard subproblems and involves critical decision-making processes. The model is governed by an objective function to be minimized, which integrates a series of adjustable weights associated with soft constraints, along with a set of hard constraints essential to ensure solution feasibility.

Minor comments
•	No details about the memetic algorithm proposed are included
•	Some details are planned to do in the future (local searches for instance)
•	No computational results have been included.

Conclusion
In our opinion, the work to be presented in a conference needs to be more mature and not just include a list of intentions or future work.

---

### Official Review · Reviewer_HnqM · 2025-03-18
**Review for Integrated Healthcare Timetabling Problem (IHTP)**

**Rating:** 3
**Confidence:** 5

**Review:**

Summary

The article tackles the challenge of integrated planning in the healthcare sector by formulating the Integrated Healthcare Timetabling Problem (IHTP) and proposing the use of a memetic algorithm to solve it. Three subproblems—operating room assignment, hospital admission planning, and nurse-to-patient room allocation—that have traditionally been treated separately are integrated into a single framework. The proposal is presented within the context of the IHTC 2024 competition and emphasizes a systematic methodology that spans from a thorough literature review to empirical solution evaluation.

The article is commendable for addressing the problem in a holistic manner by integrating aspects that are usually handled in isolation in the literature. This approach aims to better reflect the real complexity of hospital management, particularly in the face of growing demands due to an aging population and rising healthcare costs. The study is organized into well-defined phases: literature review, problem modeling, development of the memetic algorithm, experimental evaluation, and documentation of conclusions. The use of a memetic algorithm to tackle an NP-hard problem in the healthcare domain is a promising idea that aligns with existing research on complex optimization problems.

Areas for Improvement

One notable area for enhancement is the experimental results and evaluation methodology. Although the article outlines a method for evaluation, it currently lacks concrete experimental data or comparative analyses to substantiate the algorithm’s performance. Incorporating quantitative metrics, visualizations, and in-depth analyses to support the hypothesis of improved operational efficiency would significantly strengthen the study, ideally in a dedicated section that details the experiments conducted, the test instances used, the evaluation metrics, and comparisons with alternative methods.

Another suggestion involves expanding the technical description of the memetic algorithm. While the current exposition provides a general overview, it would be beneficial to include additional details—such as pseudocode, specific parameter settings, and a discussion on computational complexity—to offer clearer insight into the approach. Elaborating on the solution encoding process, the design of evolutionary operators, and the implementation of the local search procedure would further demonstrate the robustness and innovative aspects of the methodology.

A further recommendation is to integrate a comparative analysis with existing methods. By directly comparing the proposed method with other approaches or heuristics that have been applied to similar problems, the authors could more effectively highlight the relative strengths and weaknesses of their technique. Such a comparison would not only justify the choice of a memetic algorithm but also underscore its potential for enhancing operational efficiency.

Finally, it is important to clearly delineate the original contributions of this work in relation to previous studies. Providing an explicit summary that distinguishes both the theoretical and practical innovations—particularly in the areas of problem modeling and algorithm application—would enhance the overall impact and clarity of the article. Emphasizing the added value of integrating the subproblems would further clarify the unique contributions of the study.

---

### Official Review · Reviewer_xUxh · 2025-03-19
**This paper presents a research proposal focused on addressing the Integrated Healthcare Timetabling Problem (IHTP), as defined in the IHTC 2024 competition, through the application of a memetic algorithm. The project aims to develop a holistic model that integrates the scheduling of surgical cases, hospital admission planning, and the assignment of nurses to patient rooms, overcoming the limitations of fragmented approaches. The study hypothesizes that a memetic algorithm can yield high-quality solutions, enhancing operational efficiency and resource management in healthcare facilities.  The methodology includes a literature review, problem formulation, development and tuning of a memetic algorithm, evaluation and analysis of solutions, and documentation of findings. The memetic algorithm will incorporate solution encoding, initial population generation, a constructive heuristic, evolutionary operators, and local search techniques. The performance of the algorithm will be evaluated using metrics that quantify operational efficiency, costs, and overall quality, aligned with the competition's objectives.**

**Rating:** 4
**Confidence:** 3

**Review:**

The project addresses a relevant problem in healthcare resource management, the Integrated Healthcare Timetabling Problem (IHTP). The problem's formulation, which integrates operating theater scheduling, hospital admission planning, and nurse-to-patient room allocation, is complex and potentially useful.

The document is well-structured, and the study's motivation is presented clearly. However, the lack of information about the team and their experience limits the proposal's clarity. The methodology could benefit from greater specificity, but the lack of knowledge about the team hampers the evaluation of their ability to implement the proposed memetic algorithm.

Pros:

- The IHTP is an important problem in healthcare resource management.
- The application of a memetic algorithm is an interesting idea.
- Participation in IHTC 2024: Competition participation provides a relevant context.

Cons:

- The absence of details about the team's experience and expertise raises doubts about the project's feasibility.
- The absence of experimental results makes it difficult to assess the algorithm's performance.
- The lack of exploration of other potential solutions limits the work's contribution.


The project addresses a relevant problem and proposes a novel approach. However, the lack of information about the team raises doubts about the proposal's feasibility and quality. The authors are strongly encouraged to provide details about the team's experience and expertise, as well as experimental results and a more specific methodology, in future versions of the document.

---

### Decision · Program_Chairs · 2025-03-20

Accept